# Oral Epithelial Cells Distinguish between *Candida* Species with High or Low Pathogenic Potential through MicroRNA Regulation

Márton Horváth,[a] Gábor Nagy,[b] Nóra Zsindely,[a] László Bodai,[b] Péter Horváth,[d,e] Csaba Vágvölgyi,[a] Joshua D. Nosanchuk,[f,g] Renáta Tóth,[a] Attila Gácser[a,c]

[a]Department of Microbiology, University of Szeged, Szeged, Hungary
[b]Department of Biochemistry and Molecular Biology, University of Szeged, Szeged, Hungary
[c]MTA-SZTE Lendület Mycobiome Research Group, University of Szeged, Szeged, Hungary
[d]Synthetic and Systems Biology Unit, Biological Research Centre (BRC), Szeged, Hungary
[e]Institute for Molecular Medicine Finland, University of Helsinki, Helsinki, Finland
[f]Department of Medicine (Infectious Diseases), Albert Einstein College of Medicine, Bronx, New York, USA
[g]Department of Microbiology and Immunology, Albert Einstein College of Medicine, Bronx, New York, USA

Renáta Tóth and Attila Gácser share last authorship.

**ABSTRACT** Oral epithelial cells monitor microbiome composition and initiate immune response upon dysbiosis, as in the case of *Candida* imbalances. *Candida* species, such as *C. albicans* and *C. parapsilosis*, are the most prevalent yeasts in the oral cavity. Comparison of healthy oral epithelial cell responses revealed that while *C. albicans* infection robustly activated inflammation cascades, *C. parapsilosis* primarily activated various inflammation-independent pathways. In posttranscriptional regulatory processes, several miRNAs were altered by both species. For *C. parapsilosis*, the dose of yeast cells directly correlated with changes in transcriptomic responses with higher fungal burdens inducing significantly different and broader changes. MicroRNAs (miRNAs) associated with carbohydrate metabolism-, hypoxia-, and vascular development-related responses dominated with *C. parapsilosis* infection, whereas *C. albicans* altered miRNAs linked to inflammatory responses. Subsequent analyses of hypoxia-inducible factor 1$\alpha$ (HIF1-$\alpha$) and hepatic stellate cell (HSC) activation pathways predicted target genes through which miRNA-dependent regulation of yeast-specific functions may occur, which also supported the observed species-specific responses. Our findings suggest that *C. parapsilosis* is recognized as a commensal at low doses by the oral epithelium; however, increased fungal burden activates different pathways, some of which overlap with the inflammatory processes robustly induced by *C. albicans*.

**IMPORTANCE** A relatively new topic within the field of immunology involves the role of miRNAs in innate as well as adaptive immune response regulation. In recent years, posttranscriptional regulation of host-pathogenic fungal interactions through miRNAs was also suggested. Our study reveals that the distinct nature of human oral epithelial cell responses toward *C. parapsilosis* and *C. albicans* is possibly due to species-specific fine-tuning of host miRNA regulatory processes. The findings of this study also shed new light on the nature of early host cell transcriptional responses to the presence of *C. parapsilosis* and highlight the species' potential inflammation-independent host activation processes. These findings contribute to our better understanding of how miRNA deregulation at the oral immunological barrier, in noncanonical immune cells, may discriminate between fungal species, particularly *Candida* species with high or low pathogenic potential.

**KEYWORDS** *Candida*, oral epithelial cell, host-pathogen interaction, miRNA regulation

Address correspondence to Attila Gácser, gacsera@bio.u-szeged.hu.

The barrier function of epithelium is of paramount importance in maintaining homeostasis and protecting hosts against an array of injuries, including from microbes. Besides providing physical protection, cells of the epithelium produce and secrete various enzymes (e.g., lysozymes), peptides (e.g., defensins), and other small molecules (e.g., free oxygen radicals) that inhibit or kill diverse microbes (1). Epithelial cells also actively contribute to innate immune responses (2). Opportunistic pathogenic *Candida* species are members of the normal human mucosal microflora of the oral cavity, airways, intestinal tract, and genitals (3). These species primarily cause infections in immunosuppressed patients or individuals with disrupted barrier functions (4). When *Candida* cells are able to avoid or subvert host responses, serious and persistent local or systemic infections can arise (collectively also referred to as candidiasis), which includes life-threatening invasive infections (5). The most common species associated with systemic invasive candidiasis is *Candida albicans*, although the occurrence of non-*albicans Candida* (NAC) species has risen sharply in recent years, and invasive infections from NAC species are more frequent than *C. albicans* in many geographical regions (6, 7).

One of the most common forms of candidiasis is oral candidiasis, which is primarily caused by *C. albicans*, followed by *C. glabrata*, *C. parapsilosis*, *C. tropicalis*, and *C. pseudotropicalis* (8). All of these species may be present in the healthy oral mycobiome; however, their amount and diversity increase upon dysbiosis due to inflammation or cancer (9, 10). These conditions also significantly increase the risk of developing oral candidiasis. Recent cohort studies suggest that oral candidiasis occurs in ~32% of organ transplant patients (11), ~36% of diabetic patients (12), 55% of patients with radiation-induced stomatitis (13), and ~3% to 88% of individuals infected with HIV (depending on their immune status), depending on the geographical location (14).

Interactions between oral epithelial cells (ECs) and *C. albicans* are widely studied. In *C. albicans*, the most important step of the commensal-to-pathogen conversion is the yeast-to-hypha morphology shift. Hypha-associated proteins enable the fungus to acquire trace elements (e.g., iron) from ECs, attach to host cells, and invade through the epithelial barrier via induced endocytosis or active penetration (15, 16). Once adhered to the host's surface, fungal cells are recognized mainly by Toll-like receptors and C-type lectin receptors, which activate various signaling pathways (NF-κB and mitogen-activated protein kinase [MAPK] signaling). Epithelial damage also occurs, due to the secretion of various fungal enzymes or toxins, including candidalysin (17). As a result, a shift in the host biphasic MAPK signaling occurs, which discriminates between the commensal and pathogenic states of *C. albicans* (18). In contrast, relatively little is known about EC responses to NAC species, such as *C. parapsilosis*. This is important as the pathobiologies of these two species are extremely different. For example, *C. albicans* elicits an almost immediate and vigorous proinflammatory host responses, while the response evoked by *C. parapsilosis* is mild and delayed (19).

The milieu of the colonization site also seems to greatly influence the host response toward these species, given the following previous findings. (i) In contrast with *C. albicans*, *C. parapsilosis* is a common natural commensal of the human skin (20, 21). (ii) *C. parapsilosis* infrequently causes oral candidiasis (22, 23). One possible explanation for the markedly different host responses may be due to differences in posttranscriptional regulatory processes. MicroRNAs (miRNAs) are important players in fine-tuning the expression of genetic information. Recent studies demonstrate several pathogen-associated molecular pattern (PAMP)-inducible miRNAs as well as miRNAs activated by Toll-like receptor (TLR) signaling, such as miR-155, miR-132, miR-125b, or miR-146a (24), that exhibited altered expression upon bacterial or viral induction (25). Despite their confirmed relevance in host-pathogen interactions, only a few studies have analyzed miRNA profiles of host cells following *C. albicans* exposure. According to these investigations, miR-146 expression was significantly increased following β-glucan (a cell wall component of *C. albicans* cells) treatment in THP-1 cells, which resulted in the inhibition of the proinflammatory response (26). Heat-killed *C. albicans* cells were found to increase expression of five miRNAs in macrophages, including miR-155 and miR-146a,

mSystems®

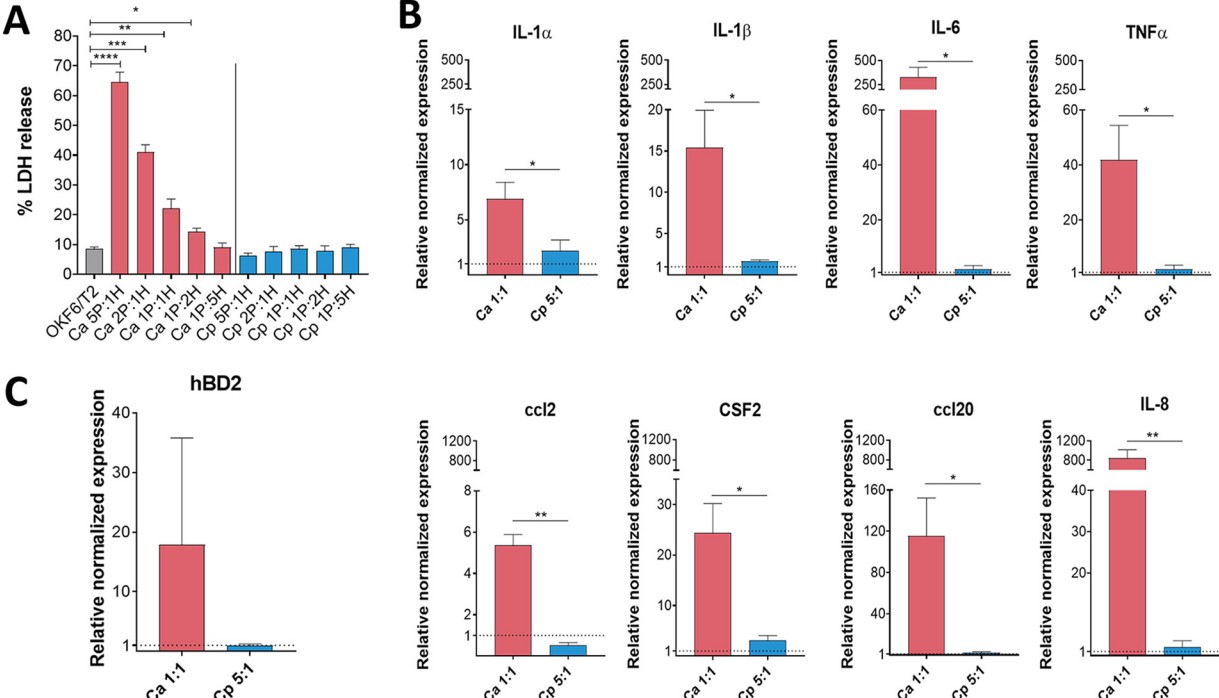

**FIG 1** Epithelial response activation by *C. albicans* and *C. parapsilosis*. OKFT6/TERT2 cells were cultured with *C. albicans* (Ca) or *C. parapsilosis* (Cp) in different infection ratios to investigate their host cell damaging capacity and EC responses after 6 h of coincubation. (A) Host cell damage was assessed by lactate dehydrogenase (LDH) measurement. (B and C) Relative normalized expression of cytokine- and chemokine-encoding genes (B) and the human β-defensin 2 (hBD2)-encoding gene (C) were determined by quantitative PCR (qPCR). The depicted significance defines the differences between the two fungal treatments. Data were normalized to the uninfected control values (set at equal to 1). Data were obtained from three independent experiments ($n = 3$) and analyzed by unpaired $t$ tests. Statistical significance is indicated by asterisks as follows: *, $P < 0.05$; **, $P < 0.01$; ***, $P < 0.001$; ****, $P < 0.0001$.

and the changes were induced by the activation of NF-κB signaling (27). In terms of epithelial barriers, the presence of *Candida*-reactive miRNAs has also been reported in airway ECs where several miRNA species associated with, for example, cell division, apoptosis, and differentiation processes, were identified (28).

In this study, we aimed to investigate how healthy oral ECs discriminate between *C. albicans* and *C. parapsilosis* and to dissect the potential underlying discriminatory mechanisms of the detected host responses. We further sought to examine whether species-specific posttranscriptional regulatory processes controlled the phenomenon by performing in-depth *in silico* analyses of both transcriptomic and miRNA sequencing data.

## RESULTS

**Robust antifungal humoral response is triggered by *C. albicans*, but not *C. parapsilosis* in oral epithelial cells.** In contrast with other innate immune cells, direct cellular responses, such as pathogen internalization and subsequent killing, are not a major function of ECs. The function of ECs manifests in the activation of professional phagocytic cells through the secretion of chemokines, cytokines, or other signaling molecules. Additional responses include the secretion of antimicrobial peptides, such as beta-defensins, another route to effectively combat invading pathogens (29, 30). To investigate the nature of healthy oral epithelial humoral responses to *C. albicans* and *C. parapsilosis*, OKF6/TERT2 cells were used. We first examined the host cell damaging capacity of both yeast species (Fig. 1A). For subsequent analyses, infection doses were selected that did not exceed 25% of host cell damage. For *C. albicans*, the multiplicity of infection (MOI) of 1:1 met this criterion (22.09% ± 3.23%), while none of the applied *C. parapsilosis* doses resulted in a more than 10% of host cell loss. Therefore, we

selected the highest infection dose (MOI of 5:1), which is in accordance with the literature (31–33).

Next, we investigated the expression of proinflammatory (tumor necrosis factor alpha [TNF-$\alpha$], interleukin 1$\alpha$/$\beta$ [IL-1$\alpha$/$\beta$], and IL-6) and immunoregulatory (granulocyte-macrophage colony-stimulating factor [GM-CSF]) cytokines, chemokines (IL-8, ccl2, and ccl20) and an antimicrobial peptide (human $\beta$-defensin 2, or hBD-2). Remarkably, *C. albicans* elevated the expression of all examined chemokines and cytokines (from 7 to 840 times higher expression) relative to the untreated control (Fig. 1B). Coculture with *C. parapsilosis* also resulted in statistically significant differences in cytokine/chemokine responses; however, compared to the exuberant immune response evoked by *C. albicans*, these changes were modest (Fig. 1B). For *C. parapsilosis*, these included IL-1$\alpha$ (2.21 $\pm$ 0.65, $P$ = 0.097), IL-1$\beta$ (1.65 $\pm$ 0.11, $P$ < 0.01), ccl2 (0.52 $\pm$ 0.1, $P$ < 0.01), and CSF2 (3.12 $\pm$ 0.53, $P$ < 0.05) relative to the untreated sample's normalized value of 1. Although not significant, the expression of hBD-2 increased in the presence of *C. albicans* only (Fig. 1C). Hence, we found marked differences in the immune response triggered by the two *Candida* species. Next, we aimed to examine whether these distinctive responses were due to alterations in regulatory processes during stimulation with these fungi.

**Species-specific host gene and miRNA expression profiles detected for *C. parapsilosis* and *C. albicans*.** Transcriptome and miRNA profile analyses were performed to further examine the distinctive responses of the ECs to the two yeast species. To obtain analysis-ready count data from the raw sequencing result files, we followed the pipeline detailed in Fig. 2A. In addition to the above-mentioned doses for the two species (MOI of 1:1 for *C. albicans* and 5:1 for *C. parapsilosis*), we also applied the 1:1 dose for *C. parapsilosis* for the subcellular analyses in order to have an equivalent ratio of host and pathogen for comparisons. Transcripts were analyzed both after an early (1-h) and a later (6-h) time point of fungal exposure to further examine both inflammatory, as well as potentially activated non-inflammation-related EC responses. These data were compared to those of the uninfected controls. Our results indicate that the majority of host cell responses to both species occurred 6 h after the start of the coculture, rather than shortly following the initial interactions (Fig. 2B). When comparing the MOI 1:1 infection doses of both species at 1 h of coculture, 50 differently expressed genes (DEGs) were identified with 46 (20 downregulated and 26 upregulated genes) occurring in *C. albicans* and 4 (all downregulated) in *C. parapsilosis*-treated cells. At the 6-h time point, 648 DEGs (348 downregulated and 259 upregulated genes) were identified in the setting of ECs with *C. albicans* and 79 (23 downregulated and 56 upregulated genes) with *C. parapsilosis*. Thus, at both time points, *C. parapsilosis* treatment effected the expression of a significantly lower number of genes at the MOI of 1:1.

Once the fungal load was increased however (MOI of 5:1), significantly more DEGs were identified at both times (83 DEGs at 1 h: 23 downregulated and 60 upregulated genes; 262 DEGs at 6 h: 55 downregulated and 207 upregulated), which exceeded the number of DEGs identified after *C. albicans* stimulus at 1 h. During the transcriptome analysis, we identified genes with species-specific expression and identical genes with similar or opposite expression patterns when comparing *C. albicans* and both ratios of *C. parapsilosis* (Fig. 2B). The identified DEGs under the different conditions are listed in Table S1 in the supplemental material.

Next, we examined the ECs' miRNA profile. miRNA analysis results revealed several miRNAs that were specifically expressed not only in the presence of the two species but also specific to the applied doses of *C. parapsilosis*. We identified 2, 2, and 3 mature miRNA transcripts at 1 h and 2, 8, and 16 at 6 h of *C. parapsilosis* MOI 1:1, MOI 5:1, and *C. albicans* MOI 1:1 stimulus, respectively (Fig. 2C). While the majority of miRNAs showed a condition-specific altered expression, miR-4464-3p showed a significantly increased expression at 1 and 6 h of *C. albicans* treatment compared to the untreated control. Of the differentially expressed miRNAs, 1, 2, and 2 target mRNAs were found at 1 h, while 12, 56, and 185 target mRNAs were identified at 6 h of *C. parapsilosis* MOI 1:1,

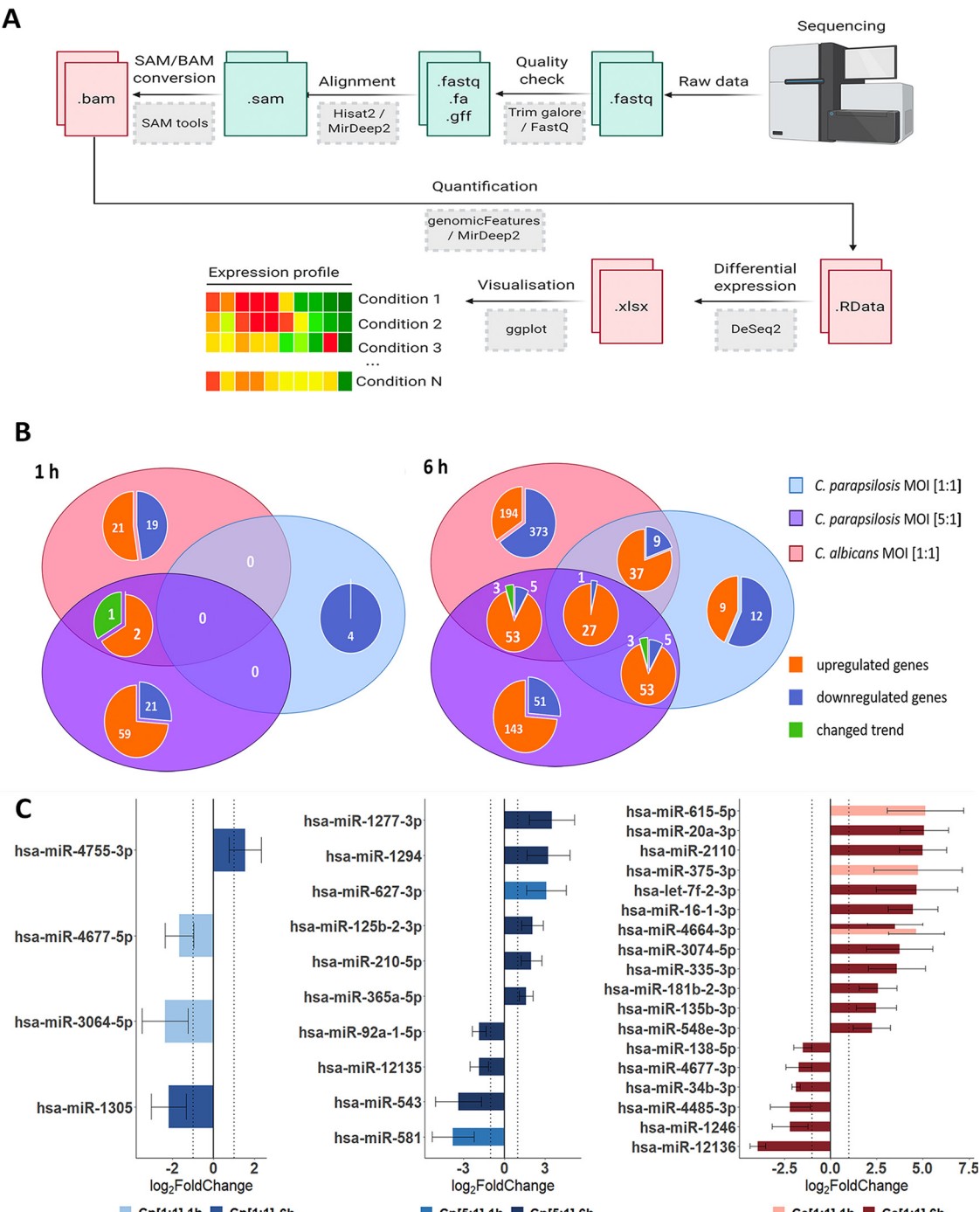

**FIG 2** Differentially expressed genes (DEGs) and dysregulated miRNAs in host responses following fungal stimuli. Host transcriptomic and miRNA responses were examined with next generation (NGS) sequencing methods (Illumina). (A) Workflow of raw data analysis, where the obtained sequences were processed alongside the above detailed bioinformatical pipeline via command line (perl- and java-based) bioinformatical tools (gray boxes), through the listed intermediate files (green/red boxes). Adapted from "Next Generation Sequencing Data Processing" by BioRender.com (2021). Retrieved from https://app.biorender.com/biorender-templates. (B) Venn diagrams of host genes identified at 1 and 6 h under each applied condition. The numbers of condition-specific genes as well as genes regulated by multiple conditions are shown. The term "changed trend" (green) refers to genes regulated by more than one condition, but the fold change was positive under at least one condition and negative in another. (C) Differently expressed host miRNA profiles after the applied conditions.

MOI 5:1, and *C. albicans* MOI 1:1 stimulus, respectively (Tables S2 to S4). These results suggest that species-specific, and in the case of *C. parapsilosis*, dose-specific posttranscriptional regulatory mechanisms regulate host responses under the applied conditions, which could explain the altered transcriptomic responses.

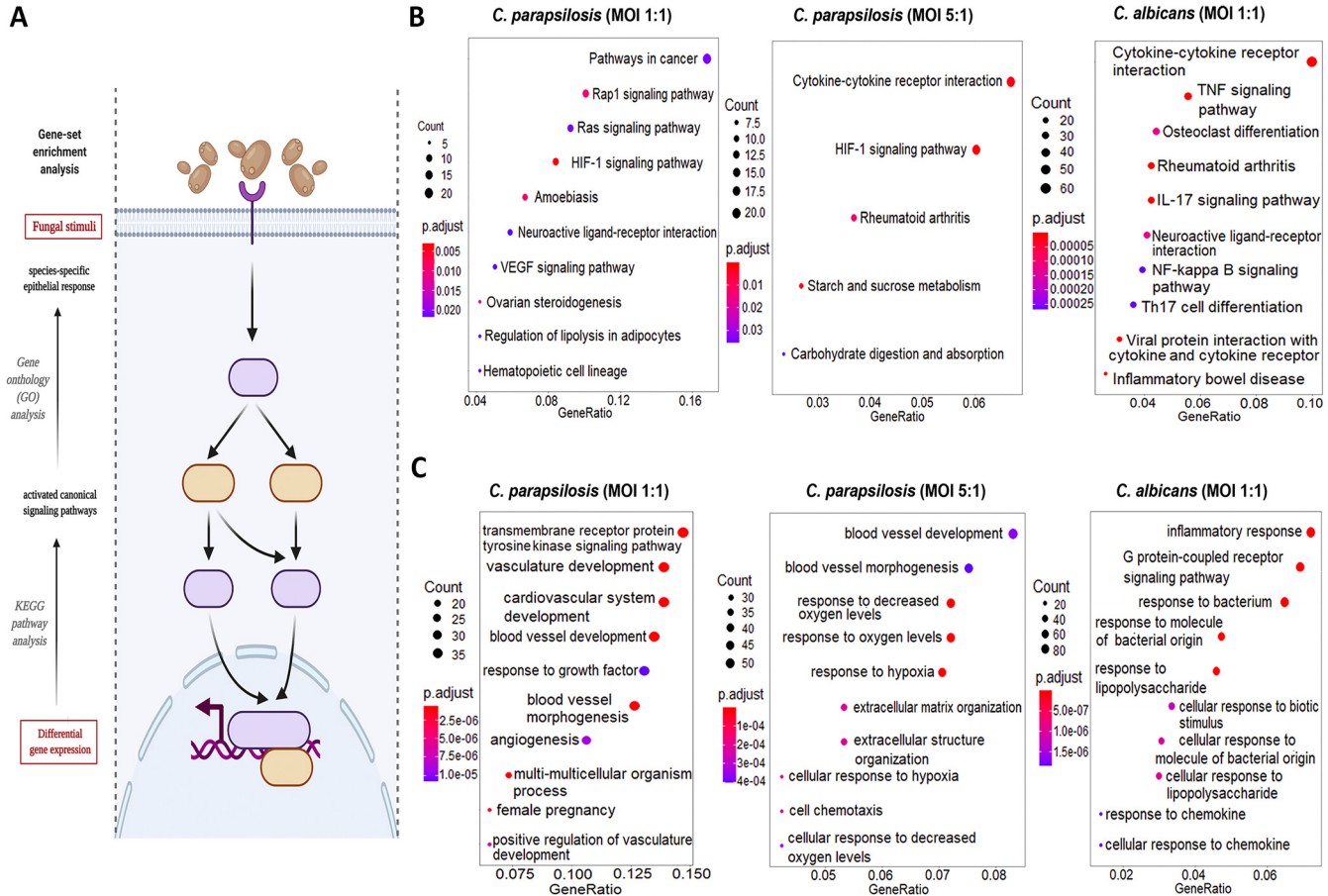

**FIG 3** Results of the KEGG pathways and GO term analyses. (A) The "enrichKEGG" and "enrichGO" functions (provided within the R-package DOSE) and "CNA" and "URA" analyses (provided within Ingenuity Pathway Analyses methods) were used to analyze the significantly up/downregulated pathways, functions, and upstream regulatory networks, respectively. (B and C) List of the 10 most activated pathways and functions after each stimuli (B). The KEGG results were labeled with their respective interest-to-background ratio (x axis on the figures) within the pathways, their significance (color coding) and with a corresponding "count," which refers to the number of DEGs within a specific pathway. Ten (or less) significant pathways with the biggest "Gene Ratio" were visualized as dotplots via the "enrichplot" package. (C) The GO term results were visualized similarly for all the applied conditions. (Created with BioRender.com.)

**Carbohydrate metabolism-, hypoxia-, and cardiovascular development-related responses dominate after *C. parapsilosis* stimulus, while *C. albicans* predominantly induces inflammation responses.** Next, we aimed to categorize the identified transcripts and characterize host responses based on the activated host signaling pathways. We employed different gene set enrichment analyses (GSEA) and overrepresentation analyses (ORA) methods (Fig. 3A) to examine the modified canonical pathways (KEGG's pathway analysis [Fig. 3B]) and biological functions based on gene ontologies (gene ontology [GO] term analysis [Fig. 3C]). At 1 h after fungal exposure, the responses detected did not allow for a more in-depth analysis. Therefore, we focused on the 6-h data set. With *C. albicans*-treated ECs, both biological pathways and functions were clearly dominated by inflammatory responses as shown by the 10 most activated pathways and functions in Fig. 3B and C. Some of the most significantly regulated pathways were the cytokine-cytokine receptor interaction, tumor necrosis factor signaling, and IL-17 signaling pathways, while activated biological functions included inflammatory responses, responses to bacteria, to molecules of bacterial origin (e.g., lipopolysaccharide [LPS]), and to chemokines (Table S5). In contrast, *C. parapsilosis* 1:1 infection resulted in the activation of routes involved in vascular development and, interestingly, pathways frequently associated with carcinogenesis (e.g., Rap1 and Ras signaling pathways, HIF1 and vascular endothelial growth factor [VEGF] signaling pathways). The affected biological pathways dominantly clustered around vascular development

(e.g., vasculature development or angiogenesis) (Table S6). Similar pathways were also activated by the *C. parapsilosis* MOI 5:1 coculture, although these were also complemented with activations of carbohydrate metabolic pathways (e.g., starch and glucose metabolism pathways) and hypoxia-related response routes (e.g., HIF1-$\alpha$ signaling pathway). Coculture of ECs with *C. parapsilosis* at an MOI of 5:1 also induced the activation of a few inflammation-related pathways (e.g., cytokine-cytokine-R interaction pathway) (Table S7). Thus, while *C. albicans* triggered multiple inflammation pathways, *C. parapsilosis* evoked a variety of mainly inflammation-independent host responses that have not been previously associated with this species.

**Potential effects of species-specific miRNA responses on host cell function.** To examine whether the identified miRNAs could affect the evoked transcriptomic responses, we first overlapped the targets of the obtained miRNAs and the corresponding altered transcriptomic profiles in each condition. Then, using cluster analyses of GO overrepresentation tests, we analyzed functions that the potential target mRNAs (or the target genes of all identified miRNAs per condition) could affect under each condition. As the early transcriptional responses under all conditions and the later transcriptional responses after *C. parapsilosis* MOI 1:1 treatment were only mild, none of the predicted functions passed the set *P* value threshold. Thus, only the differentially expressed target mRNAs derived from *C. parapsilosis* MOI 5:1 and *C. albicans* MOI 1:1 stimuli were analyzed (Fig. 4A and B). According to the GO term analyses, the top five target mRNA functions of target mRNA functions after *C. albicans* challenge were "response to bacteria," "regulation of metabolic processes," "negative regulation of cell proliferation," "response to hypoxia," and "multicellular processes' (*P* < 0.0001) (Fig. 4A and Table S8). These functions were clustered in the following categories: "response to stimulus," "metabolic processes," "cellular processes," and "developmental processes," respectively. With *C. parapsilosis* coculture, "response to hypoxia," "positive regulation of chemotaxis," "vascular processes," "positive regulation of migration," and "monosaccharide metabolic processes" were listed as the most significantly enriched functions (*P* < 0.0001), within the major identified clusters of "response to stimulus," "cell motility, "developmental processes," and "metabolic processes" (Fig. 4B and Table S9). These data suggest that the identified miRNAs could actively regulate the identified species-specific transcriptomic responses.

**HIF1-$\alpha$ pathway activation results in disrupted glucose metabolism after *C. parapsilosis* stimulus and in survival promotion after *C. albicans* infection.** In addition to the several species-specifically activated signaling pathways, we found two—HIF1-$\alpha$ and hepatic stellate cell (HSC) activation signaling pathways—that were significantly regulated in all three experimental setups at 6 h. Therefore, we examined these pathways in more depth. In the HIF1-$\alpha$ pathway, the *C. parapsilosis* MOI 1:1, MOI 5:1, and *C. albicans* MOI 1:1 stimuli resulted in the significant up- or downregulation of 7, 11, and 15 genes, respectively. In each case, we found treatment-specific activated genes as well as genes whose expression altered under at least two conditions (Fig. 5). The HIF1-$\alpha$ signaling pathway was significantly activated during all three types of stimuli compared to the "basal expression" level of the unstimulated cells. The effect was statistically most significant after the *C. parapsilosis* 5:1 treatment (*P* = 3.32e−08), followed by *C. parapsilosis* MOI 1:1 (*P* = 3.99e−07) and *C. albicans* MOI 1:1 (*P* = 1.19e−06) (Fig. 6A). We next examined potential functions that could be altered with HIF1$\alpha$ pathway deregulation. A similar activation pattern was observed in three of these biological processes: cell survival, migration, and angiogenesis. *C. albicans* clearly activated these processes (z-scores: 3.781, 3.185, and 3.481 of cell survival, migration, and angiogenesis, respectively), while the *C. parapsilosis* MOI 5:1 treatment resulted in a similar effect, but activation occurred to a lesser extent (z-scores: 3.060, 2.789, and 3.879, respectively). The *C. parapsilosis* MOI 1:1 stimulus led to only mild activation or even inhibition (z-scores of survival, 1.601, of angiogenesis, 0.860, and of migration, −0.19). Furthermore, extracellular matrix (ECM) synthesis inhibition was a characteristic of *C. albicans* treatment, while activation of glucose uptake and metabolism was a unique effect of the two *C. parapsilosis* stimuli (Fig. 6B).

## A  *C. albicans* MOI 1:1 (6 h)

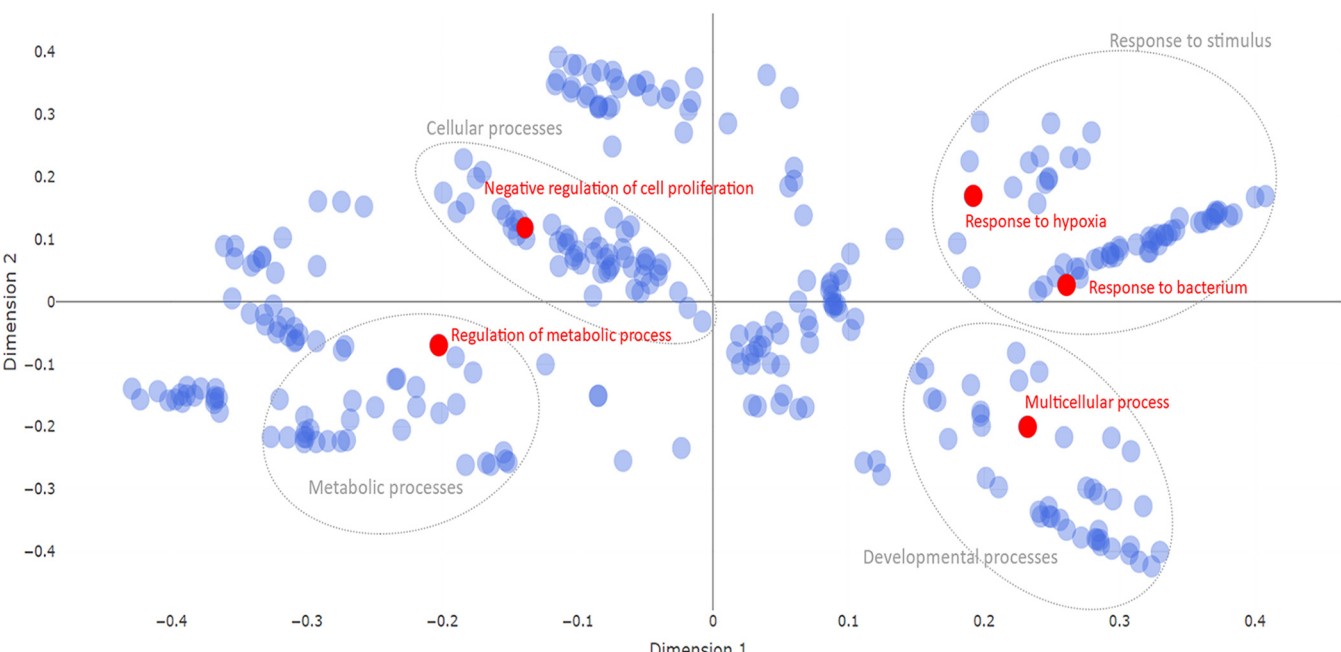

## B  *C. parapsilosis* MOI 5:1 (6 h)

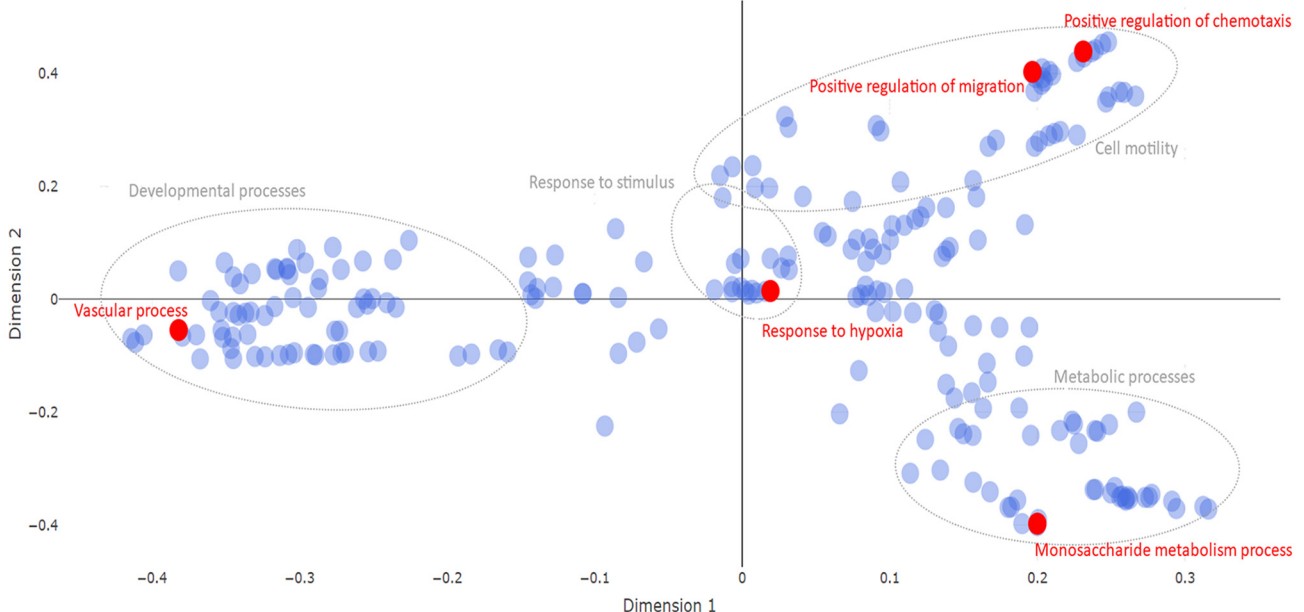

**FIG 4** Multidimensional scaling plot of target mRNA functions. Potential effects of condition-specific miRNAs on transcriptomic responses. The potential functions of target mRNA were analyzed using cluster analyses of GO overrepresentation tests. Top five functions (shown in red) of target mRNAs at 6-h *C. albicans* MOI 1:1 treatment (A) and *C. parapsilosis* MOI 5:1 challenge (B). Gray circles represent the corresponding clusters of each highlighted function.

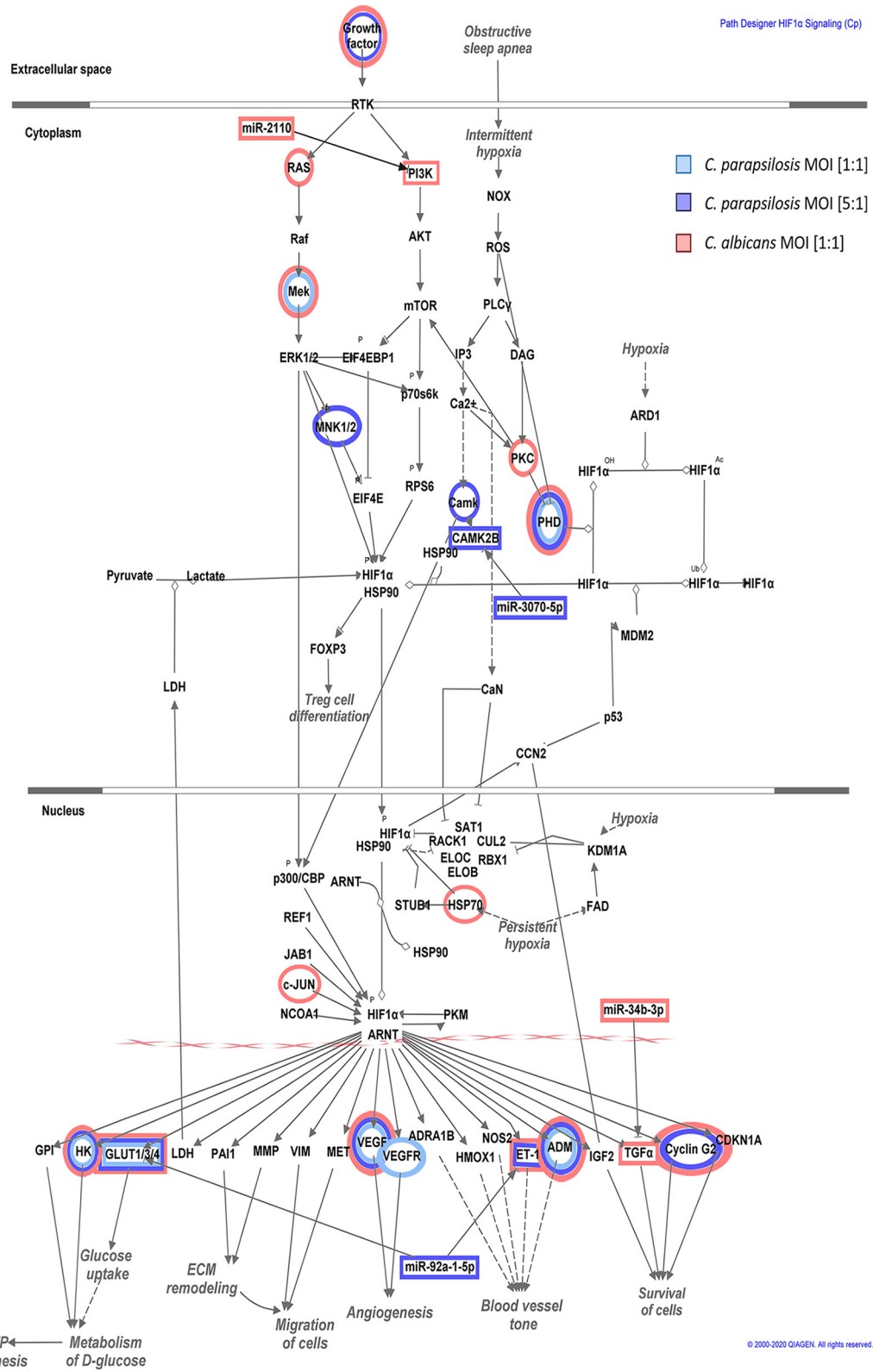

**FIG 5** Pathway explorer results on HIF1-α signaling in Ingenuity Pathway Analysis (IPA). Significantly down- or overexpressed genes in each condition were visualized within the canonical HIF1-α signal transduction network using pathway explorer and designer tools within IPA. The individual genes affected by each treatment were marked by the corresponding colors of each condition: blue for *C. parapsilosis* at an MOI of 1:1, purple for an MOI of 5:1, and red for *C. albicans* at an MOI of 1:1.

Next, we aimed to examine potential correlations between the results of the transcriptome and miRNA analyses by identifying miRNA-target mRNA pairs. Such pairs were identified after both *C. albicans* and *C. parapsilosis* MOI 5:1 coculture, but none were found for the *C. parapsilosis* MOI 1:1 condition (Fig. 6C). For *C. albicans*-treated ECs, miR-34b (downregulation, logarithmic fold change [LFC] = −1.87) and its potential target mRNA, TGF-$\alpha$ (transforming growth factor alpha; upregulation, LFC = 1.758)—a known regulator of cell proliferation and survival—was identified. Another miRNA-mRNA pair included miR-2110 (LFC = 5.00) and PIK3R3 (phosphoinositide-3-kinase regulatory subunit 3; LFC = −2.53). In the case of *C. parapsilosis* MOI 5:1-treated ECs, miR-210 (LFC = 1.98) and its potential target CAMK2B (calcium-dependent protein kinase 2 beta; LFC = −2.36) and another miRNA, miR-92a (LFC = −1.85), were identified. The latter's potential HIF1-$\alpha$ pathway target elements include EDN1 (endothelin-1 precursor; LFC = 2.11) and two glucose transporters SLC2A1 (GLUT1; LFC = 2.552) and SLC2A14 (GLUT14; LFC = 3.555) (Fig. 6C). Interestingly, an overlap could also be observed among the applied conditions in terms of the expression of specific target genes, but without the targeting miRNAs. This suggests, that under the different conditions, the expression of the examined genes is possibly regulated by other posttranscriptional regulatory processes.

**Hepatic fibrosis/stellate cell (HSC) activation pathway discriminates between the strong and attenuated inflammatory response toward the two species.** The other signaling pathway that was simultaneously regulated by all three conditions after 6 h was the hepatic fibrosis or hepatic stellate cell (HSC) pathway—a pathway involved in stellate cell activation during hepatic inflammation and injury. The early signaling events during the activation of HSCs and in activated HSCs are shown in Fig. 7. Similar to the HIF1-$\alpha$ pathway, several genes showed either species-specific or treatment-influenced expression changes (Fig. 7). The regulation of the signal transduction pathway by *C. parapsilosis* at an MOI of 1:1 and 5:1 and *C. albicans* at an MOI of 1:1 was also statistically significant, although the direction of change could not be determined, due to the incoherent changes in gene expression (Fig. 8A). This pathway is associated with a number of biological functions related to inflammation, cellular activation, chemotaxis, apoptotic cell death, or tumor cell proliferation. In general, *C. albicans* stimuli led to the overall activation of proinflammatory responses (e.g., upregulation of chemotaxis, cellular activation), while *C. parapsilosis* treatment resulted in either only a mild inflammatory response (e.g., immune cell activation with MOI of 5:1) or no significant effect (chemotaxis and cellular activation with MOI of 1:1; and inflammation and chemotaxis response for MOI of 5:1). Host cell apoptosis was inhibited by all three applied fungal conditions, although *C. parapsilosis* MOI 1:1 elicited the strongest inhibitory effect (z-score = −2.059). Interestingly, in contrast to the robust host tumor cell proliferation promoting effect of both *C. albicans* and *C. parapsilosis* at an MOI of 5:1, the low-dose application of *C. parapsilosis* led to a mild inhibitory effect (z-score = 2.260, 2.248, and −0.586, respectively) (Fig. 8B).

Similar to the HIF1-$\alpha$ pathway, miRNA-mRNA target pairs could be identified after *C. albicans* and *C. parapsilosis* MOI 5:1 stimulus, but not after the lower *C. parapsilosis* infection dose. Four miRNAs with altered expression were identified in *C. albicans*-treated cells, namely, miR-2110 (LFC = 5.00), miR-4485 (LFC = −2.19), miR-34b (LFC = −1.87), miR-4677 (LFC = −1.73), and their potential counterparts included IGFBP5 (insulin-like growth factor-binding protein 5; LFC = −3.81), KLF6 (kruppel-like factor 6; LFC = 1.58), TGF-$\alpha$ (LFC = 1.76), and IL-8 (chemokine; LFC = 9.90), respectively (Fig. 8C). After *C. parapsilosis* MOI 5:1 treatment, miR-92a and its potential target (EDN1) as well as miR-543 (LFC = −3.38) and its potential target IGFBP3 (insulin-like growth factor-binding protein 3; LFC = 4.20) were identified. Although no regulatory miRNAs were identified after *C. parapsilosis* MOI 1:1 treatment, we found altered levels of expression of KLF6 and IGFBP5, suggesting that they were possibly regulated by other posttranscriptional regulatory processes.

In this pathway, each of the targeted genes primarily affect inflammatory functions. Thus, the miRNA silencing observed here may contribute to the discrimination of the inflammatory response to *C. albicans* and *C. parapsilosis*.

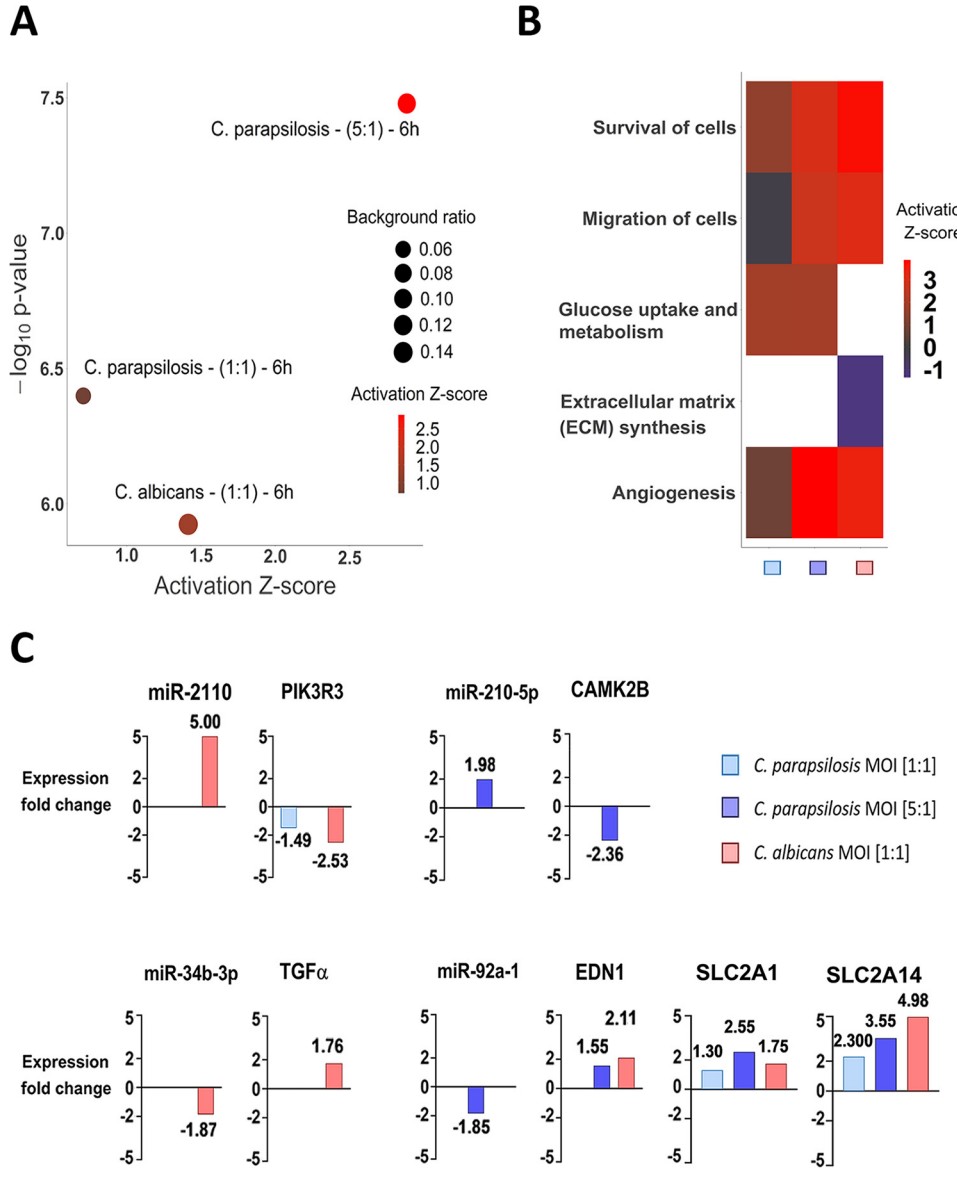

**FIG 6** Results of the IPA analyses on the HIF1-$\alpha$ signaling-related molecular components and functions. (A) The significance of the pathway activation and direction were determined by the *P* value of overlap (<0.05) after performing an expression core analysis in IPA on the set of DEGs in each applied condition. (B) The direction of activation of the functions regulated by this signaling pathway was analyzed similarly. The blank (white) rectangles mean that we could not observe a significant regulation of that particular biological function under or after the corresponding treatment. (C) We selected the miRNA-mRNA target pairs involved in this signaling pathway after applying several filtering steps in IPA's miRNA target filter tools. We only considered pairs that showed significant, opposite regulation in corresponding treatments, in which the targets were scientifically proven to be involved in HIF1-$\alpha$ signaling and the target site on the mRNA was either experimentally proven or strongly predicted by IPA based on base complementarity.

## DISCUSSION

In this study, we aimed to dissect and compare host responses triggered by *C. albicans* and *C. parapsilosis*—two common fungal residents of the oral microbial community—in oral ECs derived from a healthy individual (34). Our findings indicate that the EC immune response is more robust by 6 h of coincubation, by which time both species underwent morphology transition, rather than after 1 h, when both *C. albicans* and *C. parapsilosis* are in a yeast form or only initiating their secondary morphology, suggesting that morphology transition could be a key trigger of the epithelial cell responses. ECs actively discriminate between *C. albicans* and *C. parapsilosis*, as shown

mSystems®

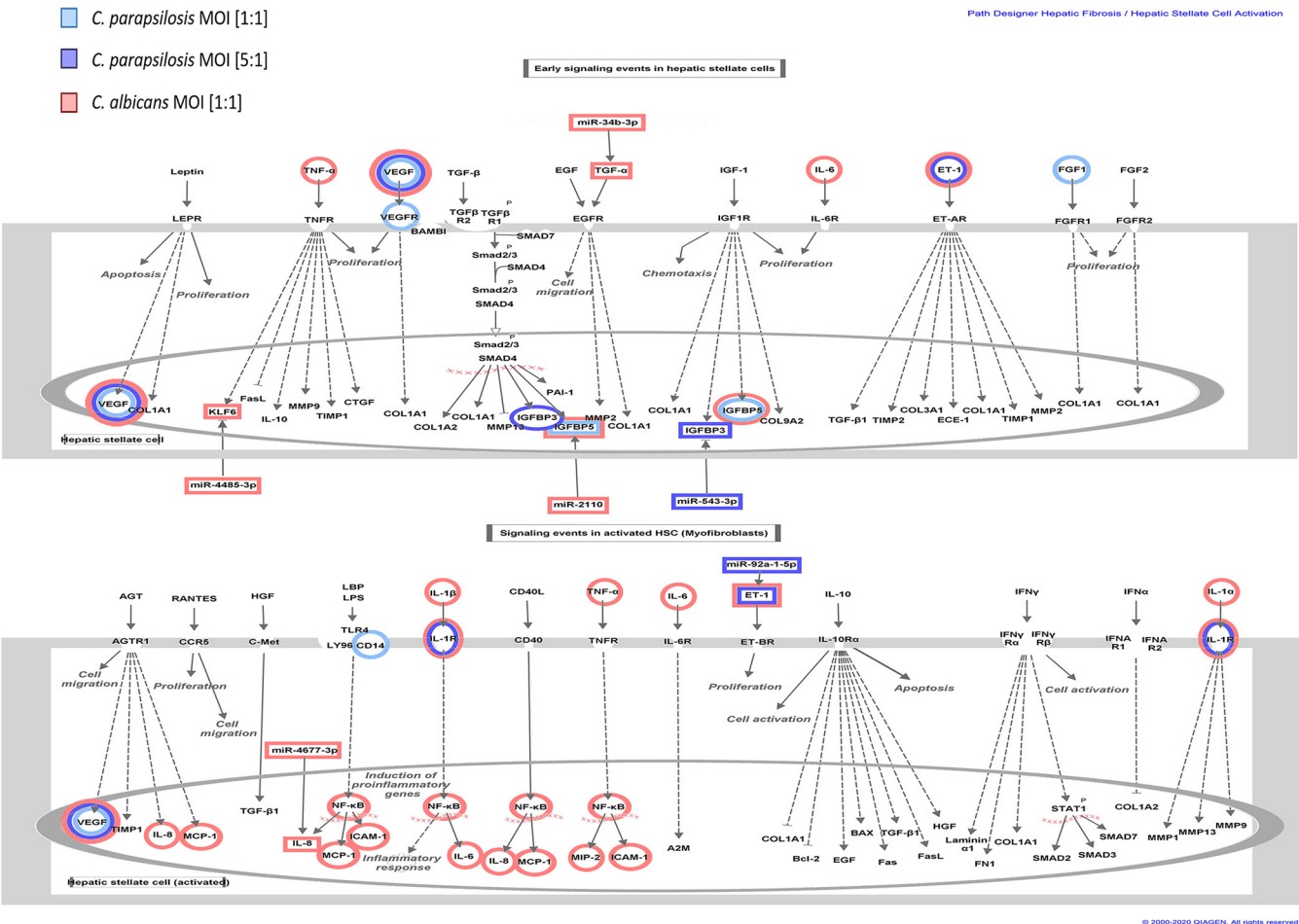

**FIG 7** Pathway explorer results on hepatic fibrosis/stellate cell activation signaling in IPA. Similar to the HIF1-α signal transduction network, regulated molecular components were visualized via pathway designer tools.

by the significant differences in host LDH release, chemokine, cytokine, and antimicrobial peptide responses. In our model, *C. parapsilosis* failed to evoke a robust, immediate proinflammatory response compared to *C. albicans*, which is similar to what has been observed in other experimental infection models (35–39). These findings are also comparable with earlier studies of *Candida*-EC interactions, showing that only *C. albicans* triggers a strong inflammatory response during the colonization of the oral epithelial barrier (36). To aid the understanding of how oral ECs might discriminate between the two species and thus distinguish a species with a higher pathogenic potential from one that more commonly is a mucosal commensal, we examined host cell transcriptomic changes following yeast-EC interaction.

Our findings revealed significant differences in host cell transcriptomic responses that were species specific. With *C. albicans* coculture, the majority of signaling routes and pathways were specific to the inflammatory response and resulted in the activation of, e.g., NF-κB and IL-17 signaling pathways (both are required for epithelium protection during oral candidiasis), which is in line with previous reports (40–42). *C. parapsilosis* challenge, however, led to the activation of various, mainly inflammation-independent pathways, such as carbohydrate metabolism-, hypoxia-, and cardiovascular development-related responses, and interestingly, pathways frequently associated with carcinogenesis, none of which have been previously associated with this species. The expression of genes related to carbohydrate metabolic processes was also upregulated in the case of both *C. parapsilosis* doses. Glucose homeostasis maintenance has recently been suggested to be required for efficient anti-*C. albicans* immune

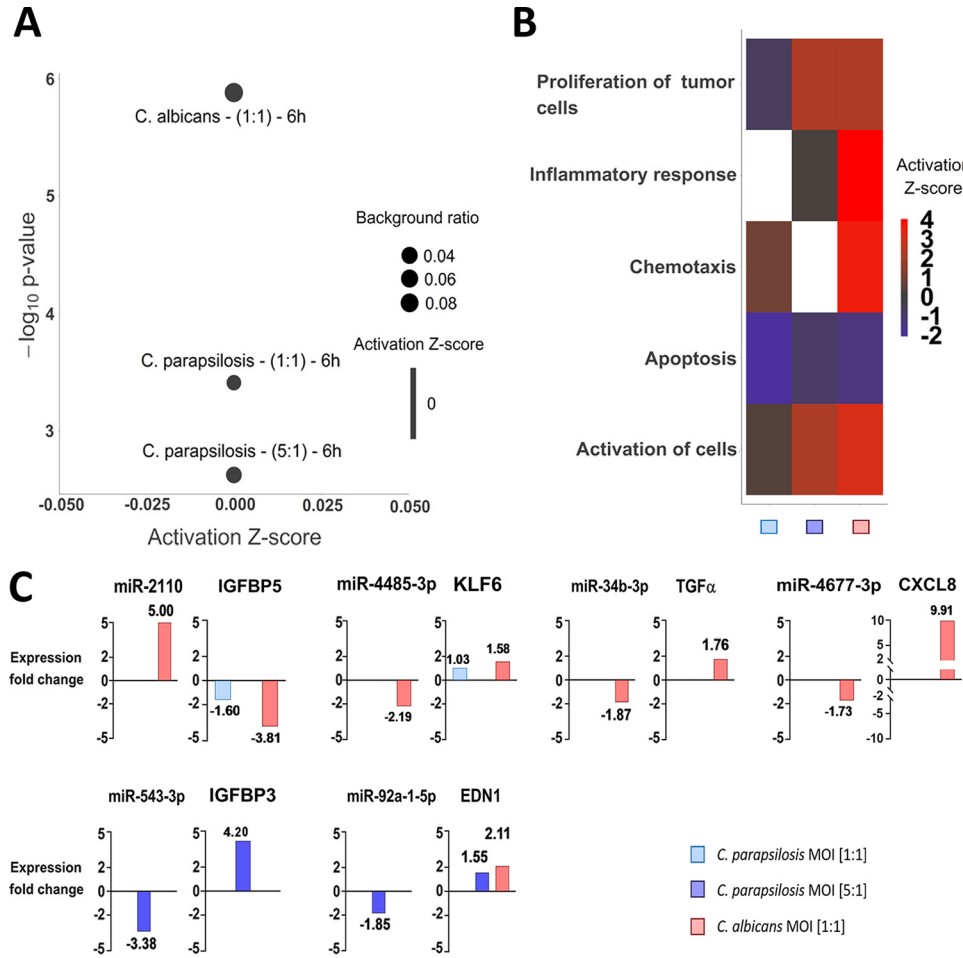

**FIG 8** Results of the IPA analyses of hepatic fibrosis/stellate cell activation signaling. (A) Direction of activation of the hepatic fibrosis signaling-related functions were analyzed after an expression core analysis of the DEGs. (B) Pathway activation was also examined. (C) miRNA-miRNA-targets were analyzed via miRNA target filter tools under similar conditions.

responses (43). According to Tucey et al (43), *C. albicans* depletes glucose from human and murine macrophages during infection, thereby accelerating host cell death. Although no studies are available comparing the carbon metabolism of *C. albicans* and *C. parapsilosis*, high glucose tolerance and rapid proliferation of *C. parapsilosis* in glucose-rich parenteral nutrition have previously been reported (44–46). This suggests enhanced glucose metabolic processes in this species, and also that during *C. parapsilosis* infections, regulation of host glucose metabolism, as a virulence factor, might be even more momentous. Although further research is required to confirm this hypothesis, considering that the highest risk group of *C. parapsilosis* infections includes low-birth-weight neonates (47), the population primarily receiving parenteral nutrition, interfering with the pathogen's carbon metabolic processes might reduce the risk of invasive candidiasis development in this patient group. Damaged tissues and inflammation are often coupled with local hypoxia (48). The lack of severe host cell damage and proinflammatory responses upon high-dose *C. parapsilosis* challenge suggests that the significantly altered hypoxic responses have other origins. Such responses could arise simply due to the elevated fungal burden rapidly depleting available oxygen levels through the rapid outgrowth of host cells. Host responses related to cardiovascular development and the activation of pathways frequently associated with carcinogenesis during *C. parapsilosis* treatment are also unique, as no such phenomenon has been previously associated with this species. Changes of expression in tumorous pathways

were dose dependent, as the low dose of *C. parapsilosis* treatment (MOI of 1:1) had no effect on expression of the related pathways, while the high-dose treatment (MOI of 5:1) significantly enhanced singling routes related to carcinogenesis, similar to that observed with *C. albicans* treatment (MOI of 1:1). Such novel information sets the ground for a new aspect of future experimental investigations in the field of *Candida* research together with cancer biology.

Besides the species-specific activated pathways, signaling pathways with simultaneous regulation by all three conditions were also found. Even among these, condition-specific transcriptional responses could be identified. One such pathway was hypoxia-inducible factor 1$\alpha$ (HIF1-$\alpha$) signaling. HIFs, especially HIF1-$\alpha$, have previously been demonstrated to regulate various innate immune processes (49). Although a study showed that HIF1-$\alpha$ activation by $\beta$-glucan and commensal bacteria promotes protection against subsequent *C. albicans* infections (50, 51), suggesting the pathway's inclusion in anti-*Candida* responses, little is known about its role in antifungal immunity regulation. Our results suggest that activation of the HIF1-$\alpha$ pathway is divergent. While *C. albicans* stimulus promoted signaling processes related to cell survival and migration, or inhibition of ECM synthesis, glucose uptake and metabolism-related processes dominated after *C. parapsilosis* coculture. While regulation of EC protective cellular responses seems to be a priority in the case of *C. albicans*, in line with previous reports (52), regulation of carbohydrate metabolism appears to be a unique characteristic of *C. parapsilosis* stimulus. The HSC activation pathway ("hepatic stellate cell activation pathway"), the other pathway simultaneously activated by all conditions, is not a singular signaling route in a strict sense, but rather a collection of several extracellular signaling molecules and their related pathways whose activation altogether lead to the transformation of stellate cells into proliferative, fibrogenic myofibroblasts under suitable conditions. These pathways include, for example, TGF-$\beta$/SMAD and TGF-$\alpha$/EGFR (epidermal growth factor receptor) pathways, which are important participants in these processes (53). These pathways are also known to influence epithelial cell functions, including their proliferation and chemotaxis (54, 55). In the HSC activation pathway, both species regulated signaling processes primarily involved in inflammation. Although *C. albicans* challenge resulted in the overall activation of proinflammatory responses, *C. parapsilosis* coculture led to only a mild effect. Species-specific regulation of both HIF1-$\alpha$ signaling and the HSC activation pathway might be what determines the outcome of the triggered innate immune responses of oral ECs.

Subsequent miRNA analyses revealed condition-specific posttranscriptional regulation of the transcriptomic responses. Among the identified dysregulated miRNA species, 4 were associated with coculturing ECs with *C. parapsilosis* at an MOI of 1:1, 10 with an MOI of 5:1, and 18 with *C. albicans*. Among the miRNAs identified during *C. albicans* treatment, only miRNA-16-1p has been previously associated with *C. albicans* infections (28). Out of the remaining 17 differentially expressed miRNAs, miR-20a and hsa-let-7 have been reported to regulate antifungal responses, although only in *Paracoccidioides brasiliensis* (56). miR-16 and miR-4677 have been linked to antibacterial host responses (57, 58), while miR-3074, miR-335, miR-34b, miR-4485, and miR-1246 have been associated with antiviral immune responses in various *in vitro* models (59–63). The remaining nine identified miRNA species have not yet been associated with microbe-induced inflammatory responses. In contrast, except for miR-3064 and miR-1294, all of the miRNAs differentially regulated in the presence of *C. parapsilosis* have been suggested to regulate host responses during microbial stimuli. miR-210 was previously associated with *C. albicans* (26), miR-125b with *P. brasiliensis* (64), and miR-92a with *Paracoccidioides americana* infections (65). Other than antifungal host responses, miR-4755 and miR-4677 deregulation was previously linked to bacterial stimuli (58, 66), miR-1305, miR-627, miR-543, and miR-581 to viral challenge, and miR-12135 to parasitic infections (67). miR-1277 and miR-365 were associated with host responses upon both bacterial and viral infections (68–72). Thus, although fewer miRNA species could be coupled with *C. parapsilosis* infections than with *C. albicans*,

the majority of these were confirmed regulators of antimicrobial responses. It is noteworthy that several of the miRNAs identified after both *C. albicans* and *C. parapsilosis* stimuli have also been associated with various tumorigenic processes (73–78), further highlighting that fungal colonization might actively influence tumorigenic processes, as suggested previously (79, 80).

Subsequent analyses revealed that the yeast-specifically-identified miRNA species regulate the expression of genes involved in condition-specific activated pathways, including survival, proliferation, and inflammation in *C. albicans* and vascular development- and carbohydrate metabolism-related pathways in *C. parapsilosis* cocultures. For instance, miR-92a was identified in HIF1-$\alpha$ signaling, potentially regulating the expression of GLUT1 (SLC2A1) and GLUT14 (SLC2A14), two glucose transporters required for carbohydrate metabolism maintenance (81) as well as the expression of EDN1 (endothelin-1) (82), a potent vasoconstrictor, during *C. parapsilosis* infection. The finding that both GLUT1 and GLUT14 are also upregulated upon *C. albicans* stimulus suggests that the significant activation of carbohydrate metabolic processes by *C. parapsilosis* is the result of an additive effect and that glucose metabolic regulators, other than the mentioned glucose transporters, are also deregulated during the stimulus. In the same pathway, following *C. albicans* challenge, miR-34b was linked to TGF-$\alpha$ expression, a known regulator of survival and cell proliferation after its activation by hypoxia-induced factors (such as HIF1-$\alpha$) (83), and miR-2110 was found to repress PIK3R3 expression, a subunit of PI3K, thereby interfering with proinflammatory responses (84, 85).

In the HSC activation pathway, besides miR-92a and its target EDN-1, miR-543 was also identified, which targets IGFBP3, an IGF-binding protein previously linked to apoptosis regulatory processes (86). Although independent of IGFBP3 expression changes, apoptosis inhibition as a potential outcome of HSC activation was predicted to be the strongest following the low-dose *C. parapsilosis* treatment. With *C. albicans* infection, besides the above-mentioned miR-34b—TGF-$\alpha$ pair, miR-2110 was identified as a potential regulator of IGFBP5, miR-4485 was linked to KLF6 regulation, and miR-467 to regulating CXCL8 expression. KLF6 is a zinc finger transcription factor previously reported to promote inflammation in macrophages (87). IGFBP5, another IGF-binding protein, was reported to be a potent chemoattractant of immune cells (88). CXCL8 is a well-known chemokine secreted by oral ECs upon *C. albicans* stimuli (89), in line with our data. Thus, all three miRNA target genes are potential inflammation regulatory components of the anti-*C. albicans* oral EC response.

Taken together, the in-depth analyses of the two simultaneously, yet diversely regulated signaling pathways also support the major, species-specific findings of the transcriptome functional analyses and suggest that the differentiating EC responses might indeed derive from altered posttranscriptional regulations. Although the obtained results shed some light on the potential underlying molecular mechanisms enabling species-specific host responses, further investigations and experimental studies are required to support these findings, such as by applying various *C. albicans* and *C. parapsilosis* clinical isolates simultaneously, to confirm our findings or further reveal potential strain-dependent differences, along with applying *in vivo* experimental setups to examine these interactions under complex oral environmental conditions.

In summary, we can conclude that human oral ECs are able to actively differentiate between *Candida* species through altered posttranscriptional regulatory processes (Fig. 9). While the presence of *C. parapsilosis* stimulus does not generate a robust inflammatory response in ECs, an elevated fungal burden can initiate inflammatory responses, albeit in a much less rapid and robust manner compared to *C. albicans*. Additionally, we found that different fungal burdens of *C. parapsilosis* led to the variable induction of generic alterations with the higher MOI inducing a broader and more significant response. The species-specific fine-tuning of both HIF1-$\alpha$ signaling and HSC activation pathways via miRNA silencing could also be a key to the distinct epithelial responses. The *in silico* data acquired through this project aid our current

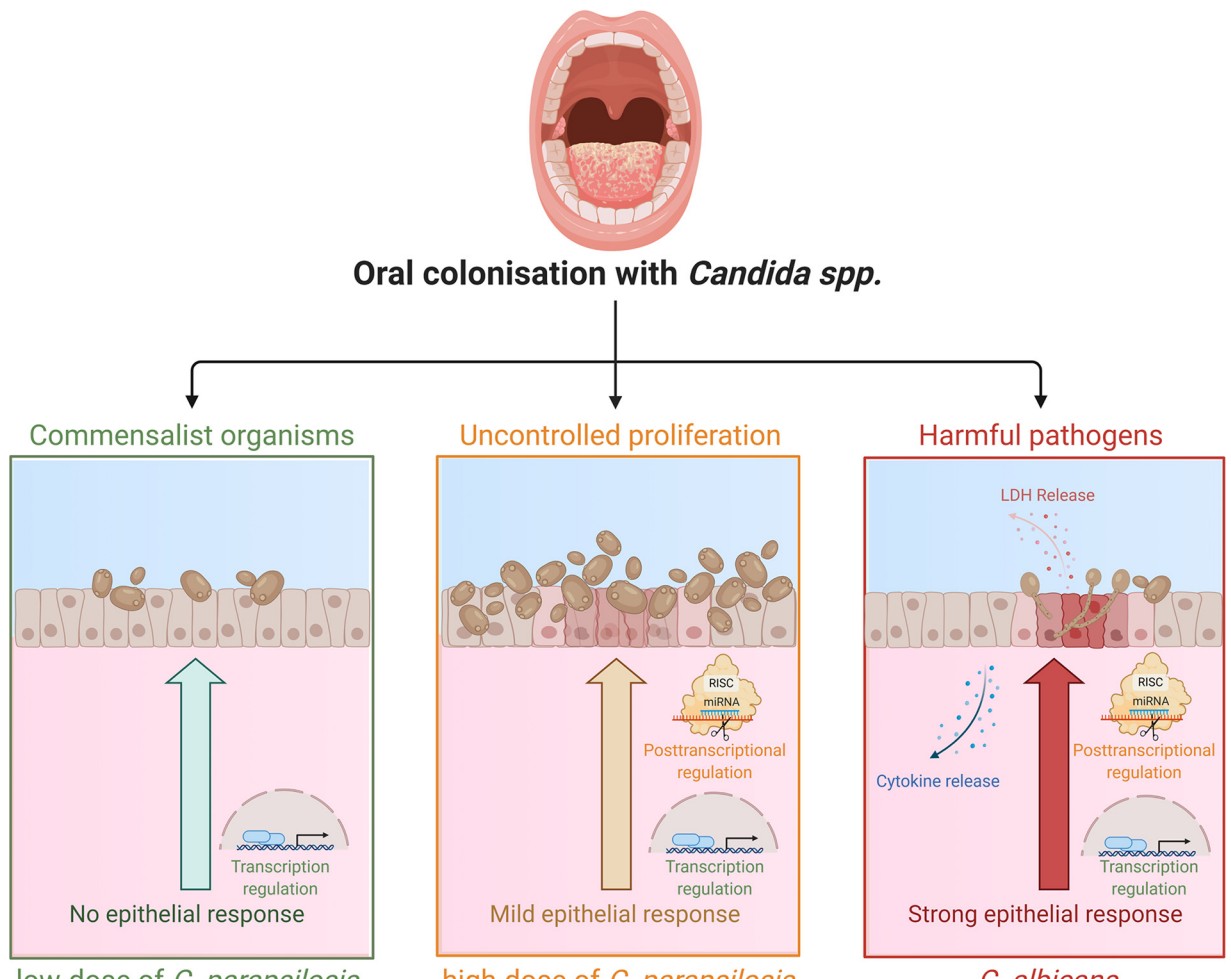

**FIG 9** Altered innate immune response regulation in healthy oral ECs discriminate between low-dose *C. parapsilosis*, increased dose of *C. parapsilosis*, and *C. albicans* stimuli. (Created with BioRender.com.)

understanding of how healthy oral ECs might discriminate between *Candida* species with high or low pathogenic potential in the human oral mucosa.

## MATERIALS AND METHODS

**Strains and growth conditions.** In this study, *Candida parapsilosis* CLIB 214 and *Candida albicans* SC5314 laboratory strains were used. *Candida* strains were maintained on solid penicillin-streptomycin-supplemented YPD medium at 4°C. Prior to host cell stimulation, yeast cells were grown overnight at 30°C in liquid yeast extract-peptone-dextrose (YPD) medium, washed three times with phosphate-buffered saline (PBS), and counted using a hemocytometer to adjust the desired cell concentration.

**Stimulation of OKF6/TERT2 cells.** The oral EC line OKF6/TERT2, a telomerase-deficient EC line derived from a healthy individual, was used for all experiments and maintained as described previously (34). Oral ECs were then plated in six-well plates in keratinocyte serum-free medium (K-SFM) supplemented with 25 µg/ml bovine pituitary extract (BPE), 2 ng/ml recombinant epidermal growth factor (rEGF), 2 mM ʟ-glutamine, and 0.5% penicillin-streptomycin, and cells were grown to 90% confluence. OKF6/TERT2 cells were then stimulated with *C. parapsilosis* and *C. albicans* in serum-free K-SFM medium. Depending on the experiment, either cell-free supernatants or host cells were collected following fungal exposure and stored at −80°C or used immediately.

**Lactate dehydrogenase assay.** Host cell damage by *C. albicans* and *C. parapsilosis* was determined by lactate dehydrogenase (LDH) cytotoxicity detection kit according to the manufacturer's instructions. OKF6/TERT2 cells were challenged with fungal cells at MOIs of 1:5, 1:2, 1:1, 2:1, and 5:1 or left untreated at various time points. During analysis, the values corresponding to the levels of LDH activity measured in untreated samples were subtracted from the values of stimulated samples. The percentage of cytotoxicity was determined as (optical density [OD] of the experimental value/OD of the positive control) × 100. 1% Triton X-100-treated samples served as positive controls. Results are derived from three independent experiments.

**Total RNA and miRNA extraction.** Total RNA and miRNA extraction from OKF6/TERT2 cells was carried out with miRNeasy minikits according to the manufacturer's instructions with minor modifications, allowing for the simultaneous extraction of total RNA and miRNA. Cells were grown until 90% confluence in tissue culturing six-well plates in supplemented K-SFM medium, washed once with PBS, and stimulated with *C. albicans* (MOI of 1:1) or *C. parapsilosis* (MOI of 1:1 and/or 5:1) in unsupplemented K-SFM medium. Following coincubation, host cells were washed two times with PBS and treated with the supplied QIAzol lysis reagent, avoiding the use of police rubber and extensive vertexing of the samples to prevent the lysis of fungal cells (hyphae). After host cell lysis, phase separation, and purification of RNA (both total and miRNA), we performed quantity and quality checks of the samples before proceeding to cDNA library preparation or cDNA synthesis and subsequent sequencing. Three independently treated biological parallels were used.

**cDNA synthesis and real-time PCR analysis.** For preliminary expression studies, 1,000 ng of RNA was utilized for cDNA synthesis using the RevertAid first strand cDNA synthesis kit. Primers for qPCR analyses are listed in Table S10 in the supplemental material. The amplification conditions were as follows: one cycle of denaturation for 3 min at 95°C; denaturation at 95°C for 10 s; 49 cycles, with 1 cycle consisting of annealing at 60°C for 30 s and elongation at 65°C for 30 s; and a final extension step at 72°C for 30 s. $\beta$2-Microglobulin was used as an internal control. Relative normalized expression values (unstimulated host cells served as controls) were calculated and presented.

**Sequencing library preparation and RNA sequencing.** miRNA sequencing libraries were prepared using NEBNext Multiplex Small RNA Library Prep Set for Illumina following the manufacturer's protocol. Libraries were size selected using AMPure XP beads and after validation with an Agilent 2100 Bioanalyzer instrument sequenced with an Illumina MiSeq DNA sequencer using Illumina MiSeq reagent kit V3-150.

**Transcriptome analysis.** We performed the preliminary quality analysis and trimming using FastQC and Cutadapt command line tools on the raw sequence files. Next, we fit the reads to the reference genome index (GRCh38) using HISAT2 (90), with the parameters –dta –non-deterministic –rna-strandness. Read numbers were calculated using the GenomicAlignments package, and differential gene expression in logarithmic fold change (LFC) was then performed using the DeSeq2 tool (91). We filtered out objects with read counts lower than 1 part per million (ppm). In the experimentally derived gene list, differentially expressed genes (DEGs) were counted above the absolute value of the logarithmic fold change of >1.5 and the adjusted *P* value of <0.05.

**Short-read mapping and counting.** The sequenced reads were mapped to known microRNA precursors, and novel sequences downloaded from miRBase (version 22) using miRDeep2.0 (92). Hits with a read count below 1 ppm were filtered out from further analysis. The distorting effect caused by the hits that were expressed at an exceptionally high level (and have the largest variance) was corrected via DeSeq2, and the *P* value was corrected by null distribution using the fdrtool package. The cutoff values for the significant hits were set at a *P* value of <0.05 and the absolute value of the logarithmic fold change of >1.5.

**Overrepresentation analyses.** Upon completion of the genome-wide RNA and miRNA expression analyses, gene expression data were interpreted using overrepresentation analyses (ORA) and gene set enrichment analyses (GSEA) provided in the Bioconductor package, DOSE (93) and clusterProfiler (94) (Fig. 3A). The two ORAs—KEGG overrepresentation test and GO overrepresentation test—as well as the GO GSEA were carried out, against a constant background, for which purpose, the human genome wide annotation package ("org.Hs.eg.db") was used (95). During both analyses, the most robust Benjamini and Hochberg (96) ("BH") method was used for the multiple comparison *P* value adjustment, and pathways *P* < 0.05 were considered significantly overrepresented. The enrichment results were visualized as dotplots via the enrichplot package. For further data mining, we calculated the semantic similarity (SS) of the found GO terms to establish connections between genes targeted by a specific miRNA via the ViSEAGO package (97). These results were visualized on a multidimensional scaling plot (MDS) that represents the distance among the set of enriched GO terms on the first two dimensions, which highlight possible clustering patterns.

**Causal analyses.** We employed causal analysis methods included in the Qiagen licensed, leading-edge bioinformatical software Ingenuity Pathway Analysis (IPA), and we ran expression core analyses on our samples. Among the included algorithms, we used downstream effect analysis (DEA) to observe each treatment's effect on the biological functions of the host cells. Furthermore, we concluded miRNA-target analyses to find possible miRNA-mRNA target pairs with significant, anti-correlated expression. We employed the *P* value of overlap and the activation Z-score to determine the significance of the prediction in IPA, which are the two most important parameters to achieve this (98). The *P* value of overlap determines the statistical significance based on the overlap of the observed and predicted regulated gene sets, while the activation Z-score predicts the direction of regulation depending on the parallelism in the observed and predicted up/downregulatory patterns. In our experiments, only the predictions with *P* values of <0.05 were considered significant hits. We further specified that only experimentally proven or strongly predicted intermolecular relationships should be considered.

**Statistical analysis.** All statistical analyses were performed with GraphPad Prism v 6.0 software using parametric *t* tests or nonparametric Mann-Whitney tests. The values for the groups examined were considered statistically significantly different at *P* < 0.05.

**Data availability.** Sequencing data are accessible under the BioProject accession number PRJNA715092.

## SUPPLEMENTAL MATERIAL

Supplemental material is available online only.

**TABLE S1**, XLSX file, 0.1 MB.

**TABLE S2**, XLSX file, 0.01 MB.

**TABLE S3**, XLSX file, 0.02 MB.
**TABLE S4**, XLSX file, 0.03 MB.
**TABLE S5**, XLSX file, 0.02 MB.
**TABLE S6**, XLSX file, 0.01 MB.
**TABLE S7**, XLSX file, 0.01 MB.
**TABLE S8**, XLSX file, 0.1 MB.
**TABLE S9**, XLSX file, 0.04 MB.
**TABLE S10**, XLSX file, 0.01 MB.

## ACKNOWLEDGMENTS

M.H. and this research work were supported by the Szeged Scientists Academy under the sponsorship of the Hungarian Ministry of Innovation and Technology (FEIF/433-4/2020-ITM_SZERZ). This work was supported by grants 20391-3/2018/FEKUSTRAT, NKFIH K 123952, and GINOP-2.3.2.-15-2016-00015. A.G. was further funded by LP2018-15/2018. László Bodai was supported by the ÚNKP-20-5-SZTE-642 New National Excellence Program of the Ministry for Innovation and Technology and by the János Bolyai Research Scholarship (BO/00522/19/8) of the Hungarian Academy of Sciences.

The human oral epithelial cell line, OKF6/TERT-2, was kindly provided by J. Rheinwald (Harvard University, Cambridge, Massachusetts).

We declare that we have no conflicts of interest.

A.G. and R.T. contributed to the concept and design of this project. M.H. and R.T. carried out the majority of experiments with the help of G.N., N.Z., C.V., J.D.N., and M.H., and G.N., L.B., and P.H. analyzed the acquired data. M.H. prepared the manuscript and the figures, which were revised by R.T. with A.G. All authors reviewed the manuscript, contributed to the discussion, and approved the final version.

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
