## [Reviewer comments · mSystems]

Oral epithelial cells distinguish between *Candida* species with high or low pathogenic potential through miRNA regulation

Marton Horvath, Gabor Nagy, Nora Zsindely, László Bodai, Peter Horvath, Csaba Vágvölgyi, Joshua Nosanchuk, Renáta Tóth, and Attila Gacser

Corresponding Author(s): Attila Gacser, University of Szeged

Review Timeline:

Submission Date:	February 11, 2021
Editorial Decision:	March 12, 2021
Revision Received:	March 23, 2021
Accepted:	April 14, 2021

Editor: Tamia Harris-Tryon

Reviewer(s): The reviewers have opted to remain anonymous.

Transaction Report:

DOI: <https://doi.org/10.1128/mSystems.00163-21>

March 12, 2021

Prof. Attila Gacser
University of Szeged
Department of Microbiology
Kozep fasor 52
Szeged 6726
Hungary

Re: mSystems00163-21 (Oral epithelial cells distinguish between *Candida* species with high or low pathogenic potential through miRNA regulation)

Dear Prof. Attila Gacser:

Below you will find the comments of the reviewers.

To submit your modified manuscript, log onto the eJP submission site at <https://msystems.msubmit.net/cgi-bin/main.plex>. If you cannot remember your password, click the "Can't remember your password?" link and follow the instructions on the screen. Go to Author Tasks and click the appropriate manuscript title to begin the resubmission process. The information that you entered when you first submitted the paper will be displayed. Please update the information as necessary. Provide (1) point-by-point responses to the issues raised by the reviewers as file type "Response to Reviewers," not in your cover letter, and (2) a PDF file that indicates the changes from the original submission (by highlighting or underlining the changes) as file type "Marked Up Manuscript - For Review Only."

Due to the SARS-CoV-2 pandemic, our typical 60 day deadline for revisions will not be applied. I hope that you will be able to submit a revised manuscript soon, but want to reassure you that the journal will be flexible in terms of timing, particularly if experimental revisions are needed. When you are ready to resubmit, please know that our staff and Editors are working remotely and handling submissions without delay. If you do not wish to modify the manuscript and prefer to submit it to another journal, please notify me of your decision immediately so that the manuscript may be formally withdrawn from consideration by mSystems.

Sincerely,

Tamia Harris-Tryon

Editor, mSystems

Journals Department
Reviewer comments:

Reviewer #1 (Comments for the Author):

This manuscript examines oral epithelial responses to *Candida*, comparing *C. albicans* and *C. parapsilosis*. The manuscript is generally easy to follow and the finding of differences for the species is interesting. The authors show that this signaling involves miRNA. It is well-written. I think it would be interesting from a fungal standpoint to see the morphologies of the fungal species with the epithelial cells at the time points analyzed, as they will likely vary in the conditions. I think discussion of this should be added as well. My main other comments are to add some background information upfront and clarify how some of the pathways are involved in epithelial function.

Abstract-some introduction to why *C. albicans* and *C. parapsilosis* were chosen and their different clinical phenotypes

Cardiovascular: Abstract and line 224 and page 12: Can this term be substituted for vascular? The "cardio" does not fit with the experimental context.

Line 92: Would include for HIV that this depends on their immune status.

Line 109: Separate this paragraph into 2 as it is quite long.

Line 111: Capitalize microRNA

Line 135-139: Since these data aren't shown, starting the paragraph with line 139 "The function of ECs..." would be easier to follow.

Line 149: Separate paragraph into 2

Figure 1: Increase the font size

Figure 1 and Figure 2: Under these conditions, are both *Candida* strains yeast at the start of the experiment? Are hyphae or pseudohyphae formed at the 1 and 6 h time points? Images of the interaction would be helpful.

Figure 3: Can the figure be increased in size (or the font) for easier reading.

Figure page 17: Can this be separated into 2? Then A could be increased in size. It is difficult to read the small font.

Line 304-323: It is not clear how the HSC pathway relates to epithelial cell responses. Provide a short description of this pathway for epithelial cells, maybe in place of the hepatic information. →

Lines 309-312: It is not clear how there is statistical significance and incoherent changes in gene expression.

Line 320: It is not clear how tumor cell proliferation relates to the fungal-epithelial interactions.

Page 20 figure: For B, should a different color assignment be given for above and below the + and - activation z-score?

Line 391: Was this linked to epithelial cells?

Discussion section: Could the carbohydrate metabolism be due to the fact the *C. parapsilosis* was at a higher MOI and thus utilizing more nutrients? The tumorigenic responses due to the higher MOI as well?

Please check the order of the tables. In my version supplemental table 10 is listed after 1 and before 2, but this may be due to the internet platform.

Reviewer #2 (Comments for the Author):

This manuscript provides a broadly defined analysis of the in vitro transcriptional response of oral epithelial cells to *C. albicans* and *C. parapsilosis*. The authors interrogate the effect of each *Candida* species independently and in parallel on both global gene expression and miRNA expression. In addition to providing an overview of genes and pathways that respond to these stimuli and contrast the very unique responses to each species, they provide additional data to support the role of miRNA changes in directing changes in gene expression and interrogate specific affected pathways to further define and augment the global gene expression data. Despite a few limitations inherent in the experimental design, the data are robust, the manuscript is well written and thorough, and the conclusions are well supported by the data presented. The findings add substantively to the growing literature in the field demonstrating that the host-pathogen relationship and responses when comparing *C. albicans* to non-*albicans* *Candida* species are fundamentally different. The following comments are provided for the authors' consideration:

1. There is increasing evidence that individual strains of both *C. albicans* and *C. parapsilosis* can behave in markedly different ways in a variety of experimental systems. Given the time and expense that would be involved in pursuing such differences, it is very reasonable to use representative laboratory strains as presented in this paper. However, the limitations should be acknowledged.

2. In a similar vein, the oral environment is remarkably complex and cannot be easily replicated in vivo. Although the data obtained from cultured epithelial cells are important and provide unique insights, the possibility that these responses may differ in vivo should be acknowledged, with influences of saliva and its biologically active components, oral microbiota, etc.

3. Minor point - Line 148 indicates that the "highest infection dose" was selected and a very reasonable rationale for this selection is provided. However, it took some time to find what MOI was used. It would be helpful to the reader to indicate that an MOI of 5:1 was used in this part of the text.

We would like to thank you for considering our research article entitled "Oral epithelial cells distinguish between *Candida* species with high or low pathogenic potential through miRNA regulation" for publication. We sincerely thank both reviewers for their time and effort to carefully evaluate our manuscript. We appreciate all of the comments and concerns that were raised during the review and hope our answers effectively address each point. We now revised and resubmitted our draft carefully considering and addressing the points raised.

Reviewer #1 (Comments for the Author):

Summary: This manuscript examines oral epithelial responses to *Candida*, comparing *C. albicans* and *C. parapsilosis*. The manuscript is generally easy to follow and the finding of differences for the species is interesting. The authors show that this signalling involves miRNA. It is well-written. I think it would be interesting from a fungal standpoint to see the morphologies of the fungal species with the epithelial cells at the time points analysed, as they will likely vary in the conditions. I think discussion of this should be added as well. My main other comments are to add some background information upfront and clarify how some of the pathways are involved in epithelial function.

Reply: We would like to thank this reviewer for their time and effort to review our results and all the constructive criticism that aided the improvement of our manuscript.

It would indeed be interesting to examine the fungal standpoint of the interaction with oral epithelial cells of healthy origin, however, we consider that this question falls beyond the scope of our investigations as we were primarily interested in host responses. For a detailed answer, please see Reply 9 ('R9'). For the other comments please see replies R2 and R12.

Comment 1: Abstract-some introduction to why *C. albicans* and *C. parapsilosis* were chosen and their different clinical phenotypes.

Reply 1: Thank you for the suggestion. We modified the abstract accordingly.

Lines 42-43: *Candida* species, such as *C. albicans* and *C. parapsilosis*, are the most prevalent yeasts in the oral cavity, thus these were selected for our experiments.

C2: Cardiovascular: Abstract and line 224 and page 12: Can this term be substituted for vascular? The "cardio" does not fit with the experimental context.

R2: Thank you for the comment. We substituted the term accordingly.

C3: Line 92: Would include for HIV that this depends on their immune status.

R3: We now modified the text accordingly.

C4: Line 109: Separate this paragraph into 2 as it is quite long.

R4: We modified the section as suggested.

C5: Line 111: Capitalize microRNA

R5: Correction is now made as suggested.

C6: Line 135-139: Since these data aren't shown, starting the paragraph with line 139 "The function of ECs..." would be easier to follow.

R6: Paragraph modified accordingly.

C7: Line 149: Separate paragraph into 2

R7: Paragraph modified accordingly.

C8: Figure 1: Increase the font size.

R8: Figure modified accordingly.

C9: Figure 1 and Figure 2: Under these conditions, are both *Candida* strains yeast at the start of the experiment? Are hyphae or pseudohyphae formed at the 1 and 6 h time points? Images of the interaction would be helpful.

R9: During the start of the experiment, both *Candida* strains are present in a yeast form. The hyphae forming ability of *C. albicans* and pseudohyphae forming ability of *C. parapsilosis* are induced primarily under conditions mimicking the host environment. In this case, in the presence of host cells and their host mimicking culturing conditions (e.g. elevated temperature, CO₂, pH, serum etc., although this is less studied in the case of *C. parapsilosis*). This phenomenon has been assessed by previous studies examining various host cell-*Candida* interactions. For instance, please see the video supplements of Lewis et al. 2012, and Toth et al. 2014, where host-pathogen interactions are studied during a 6h period using *C. albicans* (Lewis et al.) and *C. parapsilosis* cells (Toth et al.) cocultured with murine/human macrophages. Although similar events could be observed also in the presence of oral epithelial cells under the examined 6 h coinubation period, we did not find it necessary to expand upon this result, as this is a generally observed phenomenon, *C. albicans* hypha formation is considered to be one of the major factors contributing to the species virulence, while *C. parapsilosis* pseudohyphae might enhance pathogenicity mechanisms under certain conditions e.g. during penetration. In the current experimental setup both hypha and pseudohypha formation were observed during *C. albicans* and *C. parapsilosis* stimulus, respectively.

Nevertheless, we modified the discussion section as follows:

Lines 335 - 339: Our findings indicate that the EC immune response is more robust by 6 hours of co-incubation, by which time both species underwent morphology transition, rather than after 1 hour, when both *C. albicans* and *C. parapsilosis* are in a yeast form or only initiating their secondary morphology, suggesting that morphology transition could be a key trigger of the epithelial cell responses.

C10: Figure 3: Can the figure be increased in size (or the font) for easier reading.

R10: The figure is now modified accordingly.

C11: Figure page 17: Can this be separated into 2? Then A could be increased in size. It is difficult to read the small font.

R11: Thank you for the suggestion, we now separated Figure 5 and also 6 into two, and modified the text and their descriptions accordingly.

C12: Line 304-323: It is not clear how the HSC pathway relates to epithelial cell responses. Provide a short description of this pathway for epithelial cells, maybe in place of the hepatic information.

R12: *In silico* data analyses were performed with the platform provided by IPA. IPA contains a functionally assembled database of signalling components and pathways based on collective literature data. In IPA, the 'Hepatic stellate cell activation pathway' is not a singular signalling route in a strict sense, but rather a collection of several extracellular signalling molecules and their related pathways those activation altogether lead to the transformation of stellate cells into proliferative, fibrogenic myofibroblasts under suitable conditions. These pathways include, for example, pathways that are important participants in these processes (Yoshida et al. 2012, PMID: 22457652). These pathways are also known to influence epithelial cell functions, including their proliferation, chemotaxis (Valcourt et al. 2005, PMID: 15689496; Reneker et al. 1995, PMID: 7600984).

This information is now included in the discussion section in Lines 398 - 405.

C13: Lines 309-312: It is not clear how there is statistical significance and incoherent changes in gene expression.

R13: During the IPA analyses an unrelated statistical significance (p-value of overlap) and z-score were concluded. The calculation to attain the p-value does not include the direction of the regulation, solely determine significance based on the increased abundance of differentially express genes in a given subset (e.g. a signalling pathway), in contrast with the total gene set of interest. On the other hand, z-score of a given subset (pathway) is prepared

based on the statistically coherent regulation of its members, compared to a uniformly distributed null hypothesis. Thus, it is a likely scenario, that the activation of a given pathway is statistically significantly based on the changed expression pattern of its members. However, whether it results either in activation or inhibition of the pathway is not clear - statistically not significant - because of different activator and/or negative regulator components' expression changes incoherently. The principles of these computational methods are detailed in the following article (Kramer et al. 2013, PMID: 24336805)]

C14: Line 320: It is not clear how tumor cell proliferation relates to the fungal-epithelial interactions.

R14: Among the cumulative biological functions, we examined whether the differentially expressed genes contribute significantly to cell proliferation, which function characterizes HSC activation. Although IPA program determined no statistically significant differences in case of cell proliferation as a whole category, it found significant alterations in several subsets. The biggest difference among the condition was observed in case of the molecular subset related to the 'proliferation of tumor cell'. Molecules that are frequently related to tumorigenic events, suggesting that fungal exposure could also interfere with potential tumor initiatory modification of biological functions. This result and the identification of tumorous pathway-related miRNAs raised several additional questions and hypotheses, that are currently under investigations using tumor oral epithelial cells.

C15: Page 20 figure: For B, should a different color assignment be given for above and below the + and - activation z-score?

R15: Thank you. This case is a prime example of the previously mentioned scenario (please see reply 13). Although in each condition the regulation of the pathway was always significant, we could not compute a + or - z-score in any case, meaning that based on the expression profile of the components of the pathway it is not clear whether it results in an activation or inhibition of the pathway. This is the reason why no colours were applied on this figure.

C16: Line 391: Was this linked to epithelial cells?

R16: Both articles cited in line 391 suggest that the phenomenon is a monocyte-dependent effect. No information is available about epithelial cells in this manner.

C17: Discussion section: Could the carbohydrate metabolism be due to the fact the *C. parapsilosis* was at a higher MOI and thus utilizing more nutrients? The tumorigenic responses due to the higher MOI as well?

R17: The expression of genes related to carbohydrate metabolic processes also upregulated when the lower MOI of *C. parapsilosis* was used. Contrarily, no significant changes were detected in the corresponding gene's expression in case of the *C. albicans* treatment at the same infection dose (please see examples listed below). This was not highlighted on figure 2, as it lists only those pathways/functions that contained the most significant differences, however, this information is now included in the discussion section.

Lines 358 - 359: The expression of genes related to carbohydrate metabolic processes also upregulated in case of both *C. parapsilosis* doses.

Expression changes in tumorous pathways were indeed dose dependent, as the low dose of *C. parapsilosis* treatment (MOI 1:1) did not alter any of the related pathways/functions' expression, while the high dose (MOI 5:1) treatment had an effect similar to that of *C. albicans* (MOI 1:1). (Figure 8.) This observation is in line with previous findings, according to which *C. parapsilosis* elicits similar responses to *C. albicans* only upon higher fungal burden. This information has now been clarified in the discussion section as well.

Lines 379 - 382: Expression changes in tumorous pathways were dose dependent, as the low dose of *C. parapsilosis* treatment (MOI 1:1) had no effect on the related pathways expression, while the high dose (MOI 5:1) treatment significantly enhanced signaling routes related to carcinogenesis, similar to that observed with *C. albicans* treatment (MOI 1:1).

C18: Please check the order of the tables. In my version supplementary table 10 is listed after 1 and before 2, but this may be due to the internet platform.

R18: We checked the order of the tables and they should now be in the correct order.

Reviewer #2 (Comments for the Author):

This manuscript provides a broadly defined analysis of the *in vitro* transcriptional response of oral epithelial cells to *C. albicans* and *C. parapsilosis*. The authors interrogate the effect of each *Candida* species independently and in parallel on both global gene expression and miRNA expression. In addition to providing an overview of genes and pathways that respond to these stimuli and contrast the very unique responses to each species, they provide additional data to support the role of miRNA changes in directing changes in gene expression and interrogate specific affected pathways to further define and augment the global gene expression data. Despite a few limitations inherent in the experimental design, the data are robust, the manuscript is well written and thorough, and the conclusions are well supported by the data presented. The findings add substantively to the growing literature in the field demonstrating that the host-pathogen relationship and responses when comparing *C. albicans*

to non-albicans *Candida* species are fundamentally different. The following comments are provided for the authors' consideration:

Reply: We sincerely thank the reviewer for the throughout evaluation of our manuscript. It is greatly appreciated.

Comment 1: There is increasing evidence that individual strains of both *C. albicans* and *C. parapsilosis* can behave in markedly different ways in a variety of experimental systems. Given the time and expense that would be involved in pursuing such differences, it is very reasonable to use representative laboratory strains as presented in this paper. However, the limitations should be acknowledged.

Reply 1: Thank you for this comment. As the reviewer also pointed out, time and expenses prevented us to perform a throughout analysis of a strain comparison in case of both species. Our previous investigations suggest that under certain host-pathogen mimicking conditions, intraspecies differences were shown to be minor (THP-1 monocytes vs. *C. parapsilosis* strains: Toth et al, 2017, PMID: 28225025), while under different experimental set-ups, different strain-specific responses could be observed (Human PBMC-DM, Murine macrophages vs. *C. parapsilosis* strains: Toth et al. 2014, PMID: 25477874). Even among *C. albicans* strains significant differences may also be acknowledged. To assess species-specific responses, for this study, the *C. parapsilosis* CLIB 214 strain was selected, as a clinical isolate that is prone to appear in both yeast and pseudohyphal form simultaneously even at time point 0, thus representing the major distinctive feature of this species from *C. albicans*, and the *C. albicans* SC5314 strain was used, as one of the most virulent and widely applied laboratory strains used for the study of invasive candidiasis.

We now modified the discussion as follows:

Lines 465 - 471: Although the obtained results shed some light on the potential underlying molecular mechanisms enabling species-specific host responses, further investigations and experimental studies are required to support these findings, for instance by applying various *C. albicans* and *C. parapsilosis* clinical isolates simultaneously, to confirm our findings or further reveal potential strain-dependent differences, along with applying *in vivo* experimental set-ups to examine these interactions under complex oral environmental conditions.

C2: In a similar vein, the oral environment is remarkably complex and cannot be easily replicated *in vivo*. Although the data obtained from cultured epithelial cells are important and provide unique insights, the possibility that these responses may differ *in vivo* should be acknowledged, with influences of saliva and its biologically active components, oral microbiota, etc.

R2: Thank you for the comment. We now included this remark in the manuscript as mentioned in R1.

C3: Minor point - Line 148 indicates that the "highest infection dose" was selected and a very reasonable rationale for this selection is provided. However, it took some time to find what MOI was used. It would be helpful to the reader to indicate that an MOI of 5:1 was used in this part of the text.

R3: We now modified the manuscript accordingly.

April 14, 2021

Prof. Attila Gacser
University of Szeged
Department of Microbiology
Kozep fasor 52
Szeged 6726
Hungary

Re: mSystems00163-21R1 (Oral epithelial cells distinguish between *Candida* species with high or low pathogenic potential through miRNA regulation)

Dear Prof. Attila Gacser:

Your manuscript has been accepted, and I am forwarding it to the ASM Journals Department for publication. For your reference, ASM Journals' address is given below. Before it can be scheduled for publication, your manuscript will be checked by the mSystems senior production editor, Ellie Ghatineh, to make sure that all elements meet the technical requirements for publication. She will contact you if anything needs to be revised before copyediting and production can begin. Otherwise, you will be notified when your proofs are ready to be viewed.

- Minimum resolution of 1280 x 720
- .mov or .mp4. video format
- Provide video in the highest quality possible, but do not exceed 1080p
- Provide a still/profile picture that is 640 (w) x 720 (h) max

We recognize that the video files can become quite large, and so to avoid quality loss ASM suggests sending the video file via <https://www.wetransfer.com/>. When you have a final version of the video and the still ready to share, please send it to Ellie Ghatineh at eghatineh@asmusa.org.

Sincerely,

Tamia Harris-Tryon
Editor, mSystems

Journals Department
Table S10: Accept
Table S6: Accept
Table S3: Accept
Table S5: Accept
Table S4: Accept
Table S8: Accept
Table S7: Accept
Table S1: Accept
Table S9: Accept
Table S2: Accept